# JoReS-Diff: Joint Retinex and Semantic Priors in Diffusion Model for Low-light Image Enhancement

## ABSTRACT

Low-light image enhancement (LLIE) has achieved promising performance by employing conditional diffusion models. Despite the success of some conditional methods, previous methods may neglect the importance of a sufficient formulation of task-specific condition strategy, resulting in suboptimal visual outcomes. In this study, we propose JoReS-Diff, a novel approach that incorporates Retinex- and semantic-based priors as the additional multi-modal condition to regulate the generating capabilities of the diffusion model. We first leverage pre-trained decomposition network to generate the Retinex prior, which is updated with better quality by an adjustment network and integrated into a refinement network to implement Retinex-based conditional generation at both feature- and image-levels. Moreover, the semantic prior is extracted from the input image with an off-the-shelf semantic segmentation model and incorporated through semantic attention layers. By treating Retinex- and semantic-based priors as the condition, JoReS-Diff presents a unique perspective for establishing an diffusion model for LLIE and similar image enhancement tasks. Extensive experiments validate the rationality and superiority of our approach.

## CCS CONCEPTS

• **Computing methodologies** → **Image processing**.

## KEYWORDS

low-light image enhancement, multi-modal conditional diffusion model, Retinex model, semantic guidance

## 1 INTRODUCTION

Low-light photography is quite prevalent in the real world due to inherent environmental or technology restrictions. Low-light images are not only harmful for human perception but also for downstream vision tasks, such as object detection [3, 21] and semantic segmentation [6]. Thus, various methods for low-light image enhancement (LLIE) are proposed to improve the quality of low-light images.

Thanks to the development of diffusion models (DMs) [14, 37], numerous diffusion-based studies have been conducted for image restoration tasks [31, 35], with the goal of facilitating texture recovery. Previous diffusion-based image restoration methods choose to concatenate the low/high-quality images [34, 35] and extract features and priors [8, 11] as conditions. Furthermore, conditional DMs have already been introduced in LLIE tasks and thus several

Permission to make digital or hard copies of all or part of this work for personal or classroom use is granted without fee provided that copies are not made or distributed for profit or commercial advantage and that copies bear this notice and the full citation on the first page. Copyrights for components of this work owned by others than the author(s) must be honored. Abstracting with credit is permitted. To copy otherwise, or republish, to post on servers or to redistribute to lists, requires prior specific permission and/or a fee. Request permissions from permissions@acm.org.

*ACM MM, 2024, Melbourne, Australia*

© 2024 Copyright held by the owner/author(s). Publication rights licensed to ACM.
ACM ISBN 978-x-xxxx-xxxx-x/YY/MM
https://doi.org/10.1145/nnnnnnn.nnnnnnn

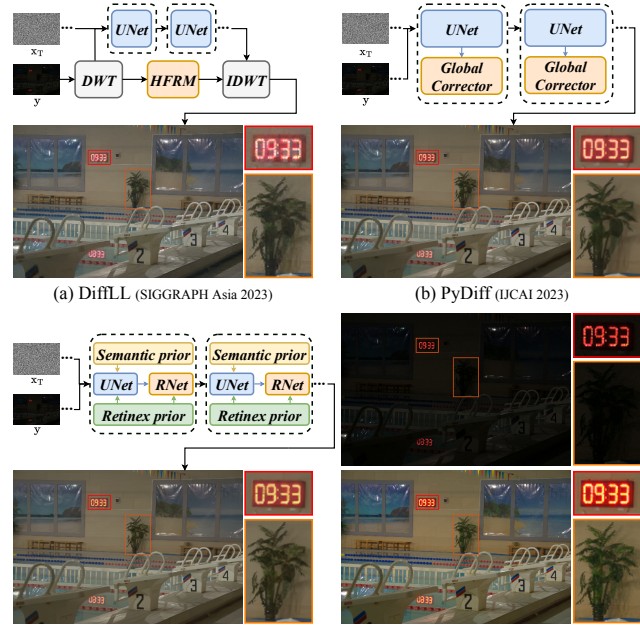

(a) DiffLL (SIGGRAPH Asia 2023)     (b) PyDiff (IJCAI 2023)

(c) JoReS-Diff (Ours)     (d) Input & GT

**Figure 1: Visual comparisons among recent DiffLL [15], Py-Diff [65], and our JoReS-Diff on LOL-v2 dataset. Previous diffusion-based methods exhibit detail loss and color distortion. Our method properly maintains color constancy and generates realistic textures thanks to the introduction of the superior Retinex and semantic priors.**

successful attempts have emerged [15, 43, 55, 56, 65]. It is worth noting that sampling efficiency is a common difficulty for the diffusion model. Therefore, Jiang *et al.* . [15] adopt 2D discrete wavelet transformations and utilize the low-resolution coefficients as conditions for faster inference speed. However, without the explicit modeling of color information, the results are unpleasing as shown in Fig. 1 (a). To better adapt to LLIE task, the color map [40, 56, 65] and Retinex model [55] are introduced into the diffusion process. [40] and [56] try to maintain color consistency, while they only apply an invariant color map and gain limited ability of enhancement. PyDiff [65] proposes a pyramid resolution setting to perform faster reverse process and adjusts the color through a global corrector, which improves the visual quality to a certain extent as shown in Fig. 1(b). However, the visual results show limitations in preserving color and details. Thus, Diff-Retinex [55] explores the possibility of establishing the Retinex-based diffusion model and produces better color and content. The denoising networks input reflectance and illumination maps as conditional images, and consistent networks are proposed to preserve the content . However, directly applying the decomposed maps as conditions is expected to be imperfect since Retinex theory is essentially an ideal model and the corruptions in reflectance and illumination need to be considered [1]. Although numerous studies have recognized the significance of

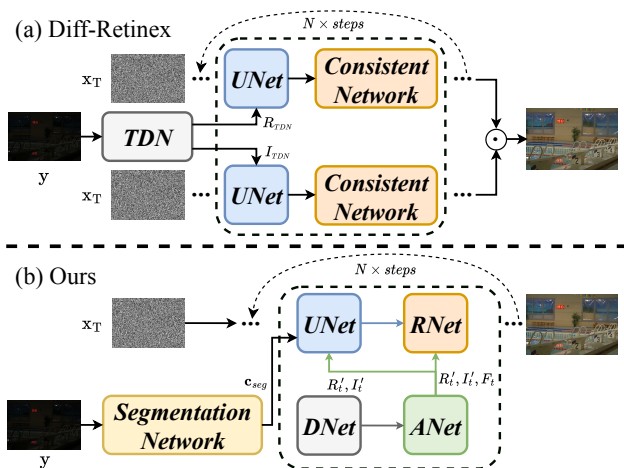

**Figure 2: Comparison between Diff-Retinex [55] and ours. Diff-Retinex still relies on the original decomposition and multiplication process uses two diffusion models to process the decomposed maps, which are multiplied as output. Our method is end-to-end and uses both Retinex and semantic priors, which are integrated into one single diffusion model. We also propose RNet to fully exploit the Retinex prior.**

auxiliary guidance in the diffusion process, they fail to explore a suitable strategy and still produce unfavorable visual results. Thus, the condition strategy has the potential for further improvement in establishing an LLIE-specific diffusion model with significant capability of enhancement.

To address the aforementioned issues, we propose a novel Retinex- and semantic-conditioned diffusion model, JoReS-Diff, specially designed for the LLIE and similar image enhancement tasks. From a new perspective, we combine physical models and semantic guidance as joint priors for the manipulation of the diffusion process. **Retinex prior:** JoReS-Diff incorporates the Retinex priors as extra pre-processing condition in the diffusion model, which is different from Diff-Retinex [55] as shown in Fig. 2. Specifically, JoReS-Diff conducts Retinex-based condition learning and conditional image refinement stages. In the former stage, we first utilize a pre-trained network (DNet) to produce initial decomposed maps. The adjustment network (ANet) suppresses noise in reflectance and adjusts exposure in illumination and outputs more reliable Retinex-based condition for better refinement. Then, JoReS-Diff incorporates the decomposed maps into the denoising U-Net (UNet) through Retinex attention layers for better conditional guidance. Furthermore, unlike [55, 65] directly adding a network after denoising process to improve image quality, we propose a refinement network (RNet) to better preserve color and contents based on the Retinex theory, which separates the illumination while maintaining the color and details in the reflectance. We reformulate the Retinex model into a residual manner to alleviate the corruptions and thus design the feature- and image-level Retinex-conditioned modules (F/IRCM) in the RNet. **Semantic prior:** Although the Retinex prior provides color and detail refinement, the diffusion model still suffers from unnatural textures [61] without considering semantic guidance. Thus, we incorporate the semantic prior for better controlling the generation ability of the diffusion model. To be specific, we extract the semantic prior from the input with an off-the-shelf semantic

segmentation model and incorporate it through semantic attention layers like [48]. We both use self-attention and cross-attention for inherent structure preservation and semantic consistency simultaneously [29]. Finally, as shown in Fig. 1(c), our JoReS-Diff provides the most pleasing result by using the novel joint condition strategy.

The main contributions of our work are as follows:

- We propose a novel diffusion-based method for image enhancement with joint Retinex and semantic priors, exploring the role of physical models and semantic guidance in controlling the generation capabilities of diffusion models.
- We propose condition learning and conditional refinement stages to integrate Retinex prior and preserve color and content consistency. We also introduce semantic prior by semantic attention layers to control the generation ability of diffusion model and reserve semantic consistency.
- Extensive experiments on representative benchmarks demonstrate that the joint Retinex and semantic priors and the well-designed interaction mechanism lead to the superior performance of our JoReS-Diff.

## 2 RELATED WORK

**Low-light Image Enhancement.** Numerous works introduce the Retinex theory into deep neural networks [1, 5, 7, 23, 26, 46, 47, 55, 60, 62]. The Retinex-Net [46] is the most inspiring method combining physical model and DNNs. Then, Zhang *et al.* . proposes KinD [62] and KinD++ [60] to provide more effective solutions. Rather than complex multi-stage training pipeline, Fu *et al.* . [5] and Cai *et al.* . [1] explore the possibility of end-to-end frameworks and achieve significant performance improvement.

Without Retinex theory, recent works concentrate on directly end-to-end manners [17, 28, 32, 42, 49–51, 58, 64]. LLNet [30] inspires the emergence of end-to-end methods. Primarily, supervised methods show promising capability of enhancement. For reducing color deviation, [22, 48, 52, 63] adopt 3DLUT and histogram to preserve color consistency. In [50, 51], the SNR-aware prior and the structure-aware features are taken as guidance to produce realistic results. Recently, the Ultra-High-Definition (UHD) images become popular. LLformer [41] and UHDFour [25] are proposed to enhance UHD images and release UHD datasets to promote the following research. Furthermore, unsupervised [7, 16] and zero-shot learning [9, 24] are valuable when training images are limited.

**Diffusion-Based Image Restoration and Low-Light Image Enhancement.** Diffusion models [14, 37] show promising capability in image generation tasks [2, 36]. For solving image processing tasks, diffusion-based methods employ conditional mechanism to incorporate the distorted images as guidance [27], such as colorization [35], super-resolution [36], restoration [4, 20, 31], and LLIE [15, 40, 43, 55, 56, 65]. To realize efficient diffusion-based LLIE, DiffLL [15] uses wavelet transformation to decrease the input size and a high-frequency restoration module to maintain the details. PyDiff [65] directly down-samples the image in early steps to speed up the sampling process. Considering the characteristics of LLIE task, LLDiffusion [40] and CLE-Diffusion [56] use the color map as extra conditional input to preserve the color information. Furthermore, PyDiff [65] proposes a global corrector to alleviate color degradation. By introducing Retinex theory, Diff-Retinex [55] decomposes the image and utilizes the reflectance and illumination maps as conditional images to form the guidance.

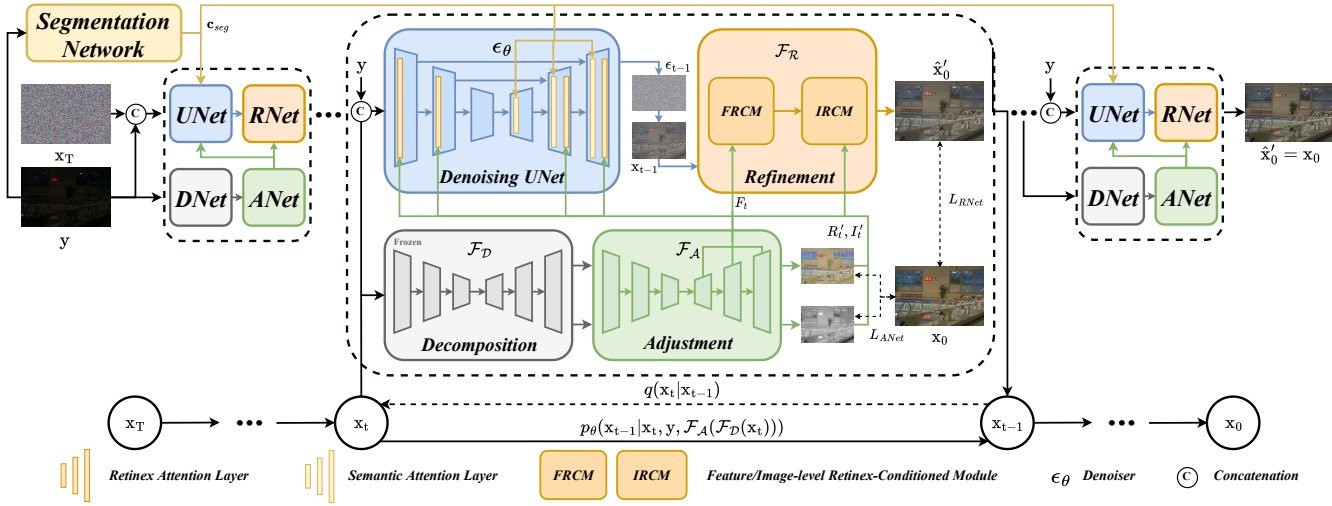

**Figure 3: Overview of our Retinex- and semantic-based conditional diffusion model (JoReS-Diff). (a) The introduction of Retinex prior contains two stages. In the learning stage, DNet provides the initial decomposed maps and ANet outputs reliable Retinex-based conditions; In the refinement stage, the conditions $R_t', L_t', F_t$ are incorporated into UNet and RNet through Retinex attention layers and F/IRCMs (detailed in Fig. 4) to preserve the color and content. (b) The semantic prior $\mathbf{c}_{seg}$ is extracted by a pre-trained segmentation model and then integrated into UNet through semantic attention layers.**

Although existing diffusion-based LLIE methods obtain good performance, they are limited by insufficient conditional guidance and produce unsatisfactory color and details. In this paper, we investigate a new condition strategy and propose JoReS-Diff to control the generative diffusion model by forming a Retinex- and semantic-based conditional process, exploring a novel perspective for diffusion-based LLIE and similar image enhancement tasks.

## 3 METHOD

The overview is shown in Fig. 3. We present the Retinex- and semantic-based condition strategy to explore an effective diffusion-based method for LLIE. We introduce the conditional DDPM in Section 3.1 and present our JoReS-Diff in Section 3.2 and Section 3.3.

### 3.1 Conditional Denoising Diffusion Model

To deal with image restoration tasks, condition strategies are proposed to develop the conditional DDPM [27]. The conditional DDPM also generates a target image $\mathbf{x}_0$ from a pure noise image $\mathbf{x}_T$ and refines the image through successive iterations. Unlike DDPM, the low-quality images are utilized as conditional inputs in conditional DDPM [36]. Thus, the conditional inference is defined as a reverse Markovian process:

$$p_\theta(\mathbf{x}_{0:T}|\mathbf{y}) = p(\mathbf{x}_T) \prod_{t=1}^{T} p_\theta(\mathbf{x}_{t-1}|\mathbf{x}_t, \mathbf{y}),$$
$$p_\theta(\mathbf{x}_{t-1}|\mathbf{x}_t, \mathbf{y}) = \mathcal{N}(\mathbf{x}_{t-1} \mid \mu_\theta(\mathbf{x}_t, \mathbf{y}, \gamma_t), \sigma_t^2 \mathbf{I}),$$
(1)

where $\gamma_t = \prod_{i=1}^{t} \alpha_i$ and $\alpha_{1:T}$ are scale parameters. Due to the absence of high-quality image $\mathbf{x}_0$ in inference, $\epsilon_\theta$ estimates noise and approximates $\mathbf{x}_0$ as:

$$\hat{\mathbf{x}}_0 = \frac{1}{\sqrt{\gamma_t}}\left(\mathbf{x}_t - \sqrt{1-\gamma_t}\,\epsilon_\theta(\mathbf{x}_t, \mathbf{y}, \gamma_t)\right).$$
(2)

Therefore, the mean of $p_\theta(\mathbf{x}_{t-1}|\mathbf{x}_t, \mathbf{y})$ can be parameterized by applying the $\hat{\mathbf{x}}_0$ into the posterior distribution [14]:

$$\mu_\theta(\mathbf{y}, \mathbf{x}_t, \gamma_t) = \frac{1}{\sqrt{\alpha_t}}\left(\mathbf{x}_t - \frac{1-\alpha_t}{\sqrt{1-\gamma_t}}\epsilon_\theta(\mathbf{x}_t, \mathbf{y}, \gamma_t)\right).$$
(3)

The training objective is to approximate the precise mean $\tilde{\mu}_\theta$, which only optimizes the denoising network $\epsilon_\theta(\cdot)$. The overall optimization objective can be formulated as:

$$\mathbb{E}_{\mathbf{x}_0, \mathbf{y}, t, \epsilon \sim \mathcal{N}(\mathbf{0}, \mathbf{I})}[\|\epsilon_t - \epsilon_\theta(\mathbf{x}_t, \mathbf{y}, \gamma_t)\|_2^2].$$
(4)

### 3.2 Retinex Prior Incorporation

Retinex theory illustrates the assumption that the reflectance and illumination components can describe the original image, and the reflectance map is consistent under various lighting conditions. Low-/normal-light images $I, \hat{I} \in \mathbb{R}^{W \times H \times 3}$ share the constant $\hat{R} \in \mathbb{R}^{W \times H \times 3}$ and consist of diverse $L, \hat{L} \in \mathbb{R}^{W \times H \times 1}$ as:

$$I = \hat{R} \odot L, \quad \hat{I} = \hat{R} \odot \hat{L},$$
(5)

where $\odot$ denotes the element-wise multiplication. According to multi-scale Retinex [19], $\hat{R}$ is the Retinex output by removing the lighting effects in $I$ as follows:

$$\hat{R} = \log(I) - \log(L) = \log(I) - \log(\mathcal{G}(I)),$$
$$\hat{I} = \mathcal{T}(\exp(\hat{R})), \quad \hat{I} \in (0, 1),$$
(6)

where $\mathcal{G}(\cdot)$ and $\mathcal{T}(\cdot)$ denote the convolution with the Gaussian surround function and the linear transformation function. However, although the ideal model maintains color constancy by estimating the reflectance map $\hat{R}$, details are easily broken through the removal of illumination. Thus, $\hat{R}$ is more suitable to guide the color recovery. We introduce the low-light input to preserve the original information and reformulate Eq. (6) to model the process as:

$$\hat{I} = \mathcal{F}(I, \hat{R}),$$
(7)

where $\mathcal{F}(\cdot)$ denotes the deep network and $\hat{R}$ acts as an auxiliary guidance. Inspired by the improved Retinex model, we consider the enhancement process from two stages: Retinex-based guidance adjustment and conditional image enhancement. Recalling the conditional DDPM model discussed in Section 3.1, the enhancement task can be performed due to the superiority of the generative model conditioned by low-light images [15, 40]. However, the noise

and artifacts in low-light images inevitably mislead the diffusion model and cause detail loss and unsatisfactory color.

To mitigate this issue, we utilize the guided enhancement manner in Eq. (7) and integrate the Retinex-based priors into the diffusion process. Notably, color constancy can actually be affected by lighting variations and the corrupted $R$ may lead to degradation of the enhanced image. Thus, we introduce the adjusted term $\hat{\mathbf{c}} = \mathcal{F}_{\mathcal{A}}(R, L)$, including more sufficient Retinex-based guidance and acting as priors in the conditional generation process. Subsequently, we formulate the enhancement model as:

$$\hat{I} = \mathcal{F}_{\mathcal{R}}(\epsilon_{\theta}(I, \mathcal{F}_{\mathcal{A}}(R, L)), \mathcal{F}_{\mathcal{A}}(R, L)), \tag{8}$$

where $\epsilon_{\theta}(\cdot)$, $\mathcal{F}_{\mathcal{R}}(\cdot)$, and $\mathcal{F}_{\mathcal{A}}(\cdot)$ denote UNet, RNet, and ANet in Fig. 3, respectively. Combining the promising texture generation capability of the diffusion model and the conscious representation with vivid color and detail of Retinex theory, we develop a Retinex-based condition strategy for the conditional DDPM and formulate our JoReS-Diff. Eq. (8) represents the denoising and refinement process in one iteration, which can be described as two stages, Retinex-based condition learning and Retinex-conditioned refinement. In the first stage, we obtain the decomposition results from a pre-trained DNet and input them into the following ANet to produce Retienx-based conditions, which will be described in Section 3.2.1. Then, the conditions will be introduced into the UNet and RNet to achieve conditional image denoising and refinement as elaborated in Section 3.2.2.

*3.2.1 Retinex-based Condition Learning.* As described in Section 3.2, the basis of conditional image refinement is to learn high-quality conditions. Inspired by Retinex-based methods [46, 62], we propose the DNet to serve as a pre-processing module and produce initial Retinex-based conditions. Compared to calculating an invariant color map as conditional input [40, 56], the decomposed maps contain not only color information but also details and illumination, achieving a more sufficient form of conditional inputs. To further exploit the progressively refined images in the diffusion process, DNet inputs $\mathbf{x}_t$ as well and provides competent updated predictions $R_t$ and $L_t$ for better generation. However, the images in the early steps contain incorrect contents and mislead the decomposition. Therefore, we use ANet to deal with corruption and ameliorate the low-quality conditions in the early steps, achieving approving conditional enhancement during the whole generative process. Thus, we realize the Retinex-based condition learning as:

$$\hat{\mathbf{c}}_t = \mathcal{F}_{\mathcal{A}}(\mathbf{c}_t) = \mathcal{F}_{\mathcal{A}}(\mathcal{F}_{\mathcal{D}}(\mathbf{y}, \mathbf{x}_t)). \tag{9}$$

**Pre-trained DNet.** DNet adopts a lightweight UNet-like network to learn the decomposition mapping. It inputs the images $I$ and outputs the reflectance $R$ and illumination $L$. Inspired by the training strategy in [46], we utilize the constant reflectance loss, smooth illumination loss, and reconstruction loss to pre-train DNet. More details can be seen in the supplementary materials.

**Condition Adjustment.** Directly using the initial decomposed maps with noise and inaccurate brightness may affect the diffusion process. Thus, we utilize the ANet to adjust the decomposition priors. The ANet employs a similar architecture to the DNet with tiny modifications on input and output layers. It inputs the initial maps $\mathbf{c}_t = [R_t, L_t]$ and learns to produce the adjusted $\hat{\mathbf{c}}_t = [R_t', L_t']$ from two aspects. We first aim to suppress the noise and calibrate the color in $R_t$ by the joint adjustment loss:

$$\mathcal{L}_{ja} = \| R_t' - \hat{R} \|_1 + SSIM(R_t', \hat{R}). \tag{10}$$

Then, we propose the joint exposure loss to adjust the sub-optimal illumination as follows:

$$\mathcal{L}_{je} = \| L_t' - \hat{L} \|_1 + Hist(L_t', \hat{L}), \tag{11}$$

where $Hist(\cdot)$ denotes the L1 loss between the histogram of $L_t'$ and ground-truth $\hat{L}$. Thus, the ANet is trained as:

$$\mathcal{L}_{ANet} = \lambda_{ja}\mathcal{L}_{ja} + \lambda_{je}\mathcal{L}_{je}, \tag{12}$$

where $\mathcal{L}_{ANet}$ is the only loss to propagate gradient for smooth optimization by detaching $R_t'$ and $L_t'$. Subsequently, ANet learns the progressive adjustment mapping of reflectance and illumination by incorporating time embedding and provides $\hat{\mathbf{c}}_t$. Furthermore, the multi-scale features $F_t$ are crucial to fully exploit the learned mapping. Thus, the features are included in conditions as $\hat{\mathbf{c}}_t = [R_t', L_t', F_t]$, providing better lightness and color guidance.

*3.2.2 Retinex-conditioned Refinement.* After obtaining the Retinex-based conditions $\hat{\mathbf{c}}_t$, the next stage is to incorporate the conditions into the iterative diffusion process. Previous methods propose to introduce the invariant color maps [40, 56, 65] and Retinex decomposition [55]. However, existing condition strategies are insufficient to conduct the favorable iterative enhancement. In our JoReS-Diff, we already learn the Retinex-based conditions as described in Section 3.2.1. Then, following the usage of color map [40, 56], we first control the generation capability of UNet by leveraging $\hat{\mathbf{c}}_t$ as extra inputs, while the carefully prepared conditions are being under-utilized just for denoising. Thus, as illustrated in Eq. (8), we apply a post-processing refinement network (RNet) to fully exploit the conditions and guarantee the consistent color and contents of $\hat{\mathbf{x}}_0'$. Different from [55], RNet adopts a lightweight architecture and implements Retinex-conditioned refinement at both feature- and image-levels by FRCM and IRCM. Overall, the refinement stage can be elaborated as:

$$\hat{\mathbf{x}}_0' = \mathcal{F}_{\mathcal{R}}(\epsilon_{\theta}(\mathbf{x}_{t-1}, \hat{\mathbf{c}}_t), \hat{\mathbf{c}}_t), \tag{13}$$

where $\hat{\mathbf{x}}_0'$ denotes the output refined by RNet at time step $t$.

**Conditional Denoising.** UNet integrates $R_t'$ and $L_t'$ by Retinex attention layers instead of input for effective interaction. We first transform $R_t'$ and $L_t'$ into embedding features $R_{em}$ and $L_{em}$. To maintain the content and color information, the reflectance embedding $R_{em}$ keeps the original resolution. The illumination embedding $L_{em}$ can be downsampled for computational efficiency thanks to the smooth distribution of lightness. Then, the $R_{em}$ and $L_{em}$ are input to Retinex attention layers with simple self-attention $F_o = SA(P_{em}, F_i)$, $P = R, L$, where $F_i, F_o$ are input and output features. Thus, we can approximate $\hat{\mathbf{x}}_0$ by updating Eq. (2) as:

$$\hat{\mathbf{x}}_0 = \frac{1}{\sqrt{\gamma_t}}\left(\mathbf{x}_t - \sqrt{1 - \gamma_t}\,\epsilon_{\theta}(\mathbf{x}_t, \mathbf{y}, \hat{\mathbf{c}}_t, \gamma_t)\right). \tag{14}$$

**Conditional Refinement.** As shown in Fig. 3, RNet consists of two Retinex-conditioned modules to realize the color recovery and detail enhancement of the approximated $\hat{\mathbf{x}}_0$. Specifically, we first implement the Retinex-conditioned refinement in feature space, since $F_t$ incorporates rich information of adjustment mapping as illustrated in Section 3.2.1. Thus, as depicted in Fig. 4, we unify the sizes of multi-scale features in $F_t$ and model them as affine transformation parameters $F_\gamma$ and $F_\beta$. Then, the feature fusion is carried out by scaling and shifting operations $\mathcal{F}_T(\cdot|\cdot)$ as:

$$F_{\hat{\mathbf{x}}_0}' = \mathcal{F}_T(F_{\hat{\mathbf{x}}_0}|F_t) = F_\gamma \odot W(F_{\hat{\mathbf{x}}_0}) + F_\beta, \tag{15}$$

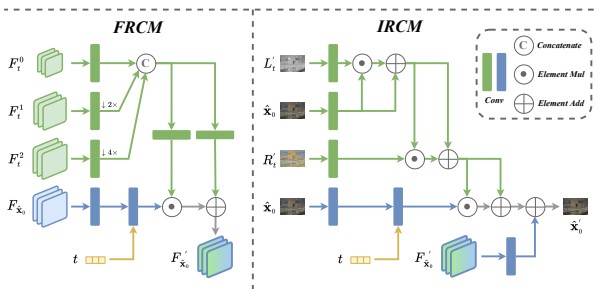

**Figure 4: The architecture of the feature/image-level Retinex-conditioned modules (F/IRCM). FRCM inputs multi-scale features $F_t$ from ANet and obtains the optimized image feature $F'_{\hat{\mathbf{x}}_0}$. Then, IRCM inputs the $R'_t$, $L'_t$ and the $F'_{\hat{\mathbf{x}}_0}$ to refine the approximated $\hat{\mathbf{x}}_0$ a produce the final output $\hat{\mathbf{x}}'_0$.**

where $W$ denotes the convolution layer and $\odot$ is the dot product. However, although the FRCM and conditional layers in UNet integrate the Retinex-based priors into the diffusion process, applying the calculation of the superior Retinex model can further refine the enhanced result. Therefore, as shown in Fig. 4, we conduct refinement at image-level by introducing $R'_t$ and $L'_t$ into IRCM. To simultaneously alleviate the errors resulting from the ideal model and establish a stable training process, we reformulate Eq. (6) into a residual refinement manner as:

$$\Delta\hat{\mathbf{x}}_0 = \mathcal{F}_T(R'_t|\mathcal{F}_T(L'_t|\hat{\mathbf{x}}_0)),$$
$$\hat{\mathbf{x}}'_0 = \mathcal{F}_T(\hat{\mathbf{x}}_0|\Delta\hat{\mathbf{x}}_0) + W(F'_{\hat{\mathbf{x}}_0}), \tag{16}$$

where $\hat{\mathbf{x}}'_0$ is the final refined output of RNet. In RNet, FRCM and IRCM fully exploit the Retinex-based conditions and produce results with consistent color and details. Accordingly, in addition to Eq. (4), we also utilize the constraint of $\hat{\mathbf{x}}'_0$ to optimize the JoReS-Diff as:

$$\mathcal{L}_{RNet} = \mathbb{E}_{\mathbf{x}_0,\mathbf{y},t,\epsilon\sim\mathcal{N}(\mathbf{0},\mathbf{I})}\left[\|\mathbf{x}_0 - \mathcal{F}_R(\hat{\mathbf{x}}_0, \hat{\mathbf{c}}_t, \gamma_t)\|_2^2\right]. \tag{17}$$

### 3.3 Semantic Prior Incorporation

Apart from the capability of color recovery and detail maintenance brought by the Retinex prior, we introduce the semantic prior to further control the generative diffusion model at semantic-level and alleviate the unnatural textures in the output image. Following the semantic-aware framework in [48], we utilize the same semantic segmentation model to produce available semantic prior from the input image. The segmentation model inputs the low-light image and outputs the intermediate features and semantic map. However, the low-light image remains the distribution gap comparing to the normal-light image, which may cause the misclassification and thus produce unsatisfactory segmentation results. Directly applying the semantic map will mislead the diffusion model in some cases, resulting in more unnatural textures. Therefore, we only use the intermediate latent features since the misclassification can be smoothed in the high-dimensional feature space. Consequently, we collect the multi-scale features from the segmentation model as the semantic prior, $\mathbf{c}_{seg} = [F_s{}^0, F_s{}^1, F_s{}^2]$. The three semantic features ($F_s^b, b = 0, 1, 2$) with three spatial resolutions ($H/2^{4-b}, W/2^{4-b}$), where $H$ and $W$ are the height and width of the input image.

Then, we propose the semantic attention layers to conduct the introduction of the semantic prior, as shown in Fig. 3. Similar to the Conditional Denoising in Section 3.2.2, we exploit the semantic

information to manipulate the feature in the decoder of the UNet as well. Notably, we design the semantic attention layer based on both the self-attention $SA(\cdot)$ and cross-attention $CA(\cdot)$. Primarily, self-attention plays a crucial role in preserving the geometric and shape details and the cross-attention contributes more to generate semantic consistency [29]. And we apply learnable weights to achieve an adaptive addition of the outputs from self- and cross-attention branches. The overall calculation can be described as:

$$F_o = \lambda_{SA}SA(F_s, F_i) + \lambda_{CA}CA(F_s, F_i), \tag{18}$$

where $\lambda_{SA}, \lambda_{CA}$ denote learnable weights, $F_i, F_o$ denote input and output features. Consequently, Eq. (4) can be reformulated as:

$$\mathcal{L}_{UNet} = \mathbb{E}_{\mathbf{x}_0,\mathbf{y},t,\epsilon\sim\mathcal{N}(\mathbf{0},\mathbf{I})}\left[\|\epsilon_t - \epsilon_\theta(\mathbf{x}_t, \mathbf{y}, \hat{\mathbf{c}}_t, \mathbf{c}_{seg}, \gamma_t)\|_2^2\right],$$
$$\mathcal{L}_{ALL} = \lambda_{UNet}\mathcal{L}_{UNet} + \lambda_{RNet}\mathcal{L}_{RNet} + \lambda_{ANet}\mathcal{L}_{ANet} \tag{19}$$

where $\lambda_s$ denote loss weights.

## 4 EXPERIMENTS

### 4.1 Experimental Settings

**Implementation Details.** Our method is trained using an NVIDIA RTX A100 GPU with 600k iterations. We use Adam optimizer with the momentum as $(0.9, 0.999)$. The initial learning rate is set to $1\times10^{-4}$ and decays by a factor of 0.5 after every $1\times10^5$ iterations. The batch size and patch size are set to 8 and 256×256. We use the similar architecture of [36] as our denoising U-Net $\epsilon_\theta$. In the inference stage, we follow the DDIM [37] and output by 8 steps.

**Datasets.** We evaluate our method on several datasets. The LOL dataset [46] includes 485 low/normal-light training pairs and 15 testing pairs. The LOL-v2 dataset [54] has two parts, the real part is an extension of LOL with 689/100 pairs and the synthetic part includes 900/100 pairs for training and testing. The UHD-LL dataset [25] includes 2000 training and 150 testing pairs. The ISTD dataset [39] includes 1330 training and 540 testing triplets. The MIT-Adobe FiveK dataset [38] includes 4500 training and 500 testing pairs.

**Metrics.** We employ peak signal-to-noise ratio (PSNR), structural similarity (SSIM) [44], perceptual image patch similarity (LPIPS) [59], and Fréchet Inception Distance (FID) [13] for evaluation.

**Compared Methods.** We compare with a rich collection of state-of-the-art LLIE methods, including LIME [12], RetinexNet [46], KinD [62], DRBN [53], Zero-DCE [9], EnlightGAN [16], MIRNet [58], SNR [50], PairLIE [7], SMG [51], FourLLIE [25], Retinexformer [1], UHDFour [25] and diffusion-based LLIE methods including Diff-Retinex [55], DiffLL [15], and PyDiff [65]. For the shadow removal and exposure adjustment tasks, we select DC-ShadowNet [18], EM-Net [67], BMNet [66], ShadowFormer [10], LFG-Diff [33], Uformer [45], Restormer [57], LLFormer [41].

### 4.2 Quantitative Evaluation

Tables 1 and 2 show the comparisons on LOL, LOL-v2, and UHD-LL. It is clear that our JoReS-Diff achieves consistent and significant performance gain over all competing methods. Specifically, our method provides significant improvement of 0.155 dB/0.216 dB on LOL/LOL-v2-real datasets respectively, establishing the new state-of-the-art with PSNR values of 26.491 dB/24.836 dB. Furthermore, our method achieves similar performance in terms of SSIM, yielding the best values of 0.876/0.897. On the LOL-v2-synthetic test set, our JoReS-Diff improves the PSNR by 0.042 than the second-best method. On UHD-LL, we also achieve considerable performance

**Table 1: Quantitative comparison on the LOL [46] and LOL-v2-real [54] datasets. ↑ (↓) denotes that, larger (smaller) values suggest better quality. The best results are highlighted in bold and the second best results are in underline (Special fonts are also used in Table 2). Note that the absent results ("-") of Diff-Retinex due to the lack of code.**

| Methods | LOL | | | | LOL-v2-real | | | | LOL-v2-synthetic | | | |
|---|---|---|---|---|---|---|---|---|---|---|---|---|
| | PSNR ↑ | SSIM ↑ | LPIPS ↓ | FID ↓ | PSNR ↑ | SSIM ↑ | LPIPS ↓ | FID ↓ | PSNR ↑ | SSIM ↑ | LPIPS ↓ | FID ↓ |
| LIME [12] TIP'16 | 16.760 | 0.560 | 0.350 | 117.892 | 15.240 | 0.470 | 0.428 | 118.171 | 16.880 | 0.758 | 0.104 | - |
| RetinexNet [46] BMVC'18 | 16.770 | 0.462 | 0.474 | 113.699 | 18.371 | 0.723 | 0.365 | 133.905 | 16.551 | 0.652 | 0.379 | 98.843 |
| KinD [62] MM'19 | 20.870 | 0.799 | 0.207 | 104.632 | 17.544 | 0.669 | 0.375 | 137.346 | 18.956 | 0.801 | 0.262 | 89.156 |
| DRBN [53] CVPR'20 | 19.860 | 0.834 | 0.155 | 75.359 | 20.130 | 0.830 | 0.147 | 60.631 | 21.687 | 0.825 | 0.174 | 52.972 |
| Zero-DCE [9] CVPR'20 | 14.861 | 0.562 | 0.335 | 101.237 | 18.059 | 0.580 | 0.313 | 91.939 | 17.756 | 0.814 | 0.168 | 49.239 |
| MIRNet [58] PAMI'22 | 24.140 | 0.842 | 0.131 | 69.179 | 20.357 | 0.782 | 0.317 | 49.108 | 21.941 | 0.876 | 0.112 | 38.775 |
| SNR [50] CVPR'22 | 24.608 | 0.840 | 0.151 | 55.121 | 21.479 | 0.848 | 0.157 | 54.532 | 24.130 | 0.927 | **0.032** | 23.971 |
| PairLIE [7] CVPR'23 | 19.510 | 0.736 | 0.248 | 100.715 | 20.357 | 0.782 | 0.317 | 96.911 | 19.074 | 0.794 | 0.230 | 85.209 |
| SMG [51] CVPR'23 | 23.684 | 0.826 | 0.118 | 58.846 | 24.620 | 0.867 | 0.148 | 78.582 | 25.618 | 0.905 | 0.053 | 23.210 |
| FourLLIE [25] MM'23 | 20.222 | 0.766 | 0.250 | 91.793 | 22.340 | 0.847 | **0.051** | 89.334 | 24.649 | 0.919 | 0.039 | 26.351 |
| Retinexformer [1] ICCV'23 | 25.153 | 0.843 | 0.131 | 71.148 | 22.794 | 0.839 | 0.171 | 62.439 | 25.670 | 0.928 | 0.059 | 22.781 |
| Diff-Retinex [55] ICCV'23 | 21.981 | 0.863 | **0.048** | 47.851 | - | - | - | - | - | - | - | - |
| DiffLL [15] SIGGRAPH Asia'23 | 26.336 | 0.845 | 0.217 | 48.114 | 22.428 | 0.817 | 0.191 | 59.075 | 25.456 | 0.896 | 0.102 | 43.670 |
| PyDiff [65] IJCAI'23 | 25.643 | 0.849 | 0.142 | 69.784 | 23.441 | 0.833 | 0.208 | 71.538 | 25.126 | 0.917 | 0.098 | 29.361 |
| **JoReS-Diff (Ours)** | **26.491** | **0.876** | 0.092 | **43.596** | **24.836** | **0.897** | 0.109 | **46.938** | **25.712** | **0.928** | 0.058 | **22.776** |

**Table 2: Quantitative comparison on the UHD-LL [25]. To conserve space, we select recent methods in 2023 while ensuring that the results are sufficient to demonstrate our superiority.**

| Methods | UHD-LL | | | |
|---|---|---|---|---|
| | PSNR ↑ | SSIM ↑ | LPIPS ↓ | FID ↓ |
| SMG [51] CVPR'23 | 25.852 | 0.869 | 0.248 | 41.647 |
| FourLLIE [25] MM'23 | 22.462 | 0.814 | 0.296 | 61.380 |
| UHDFour [25] ICLR'23 | 26.226 | 0.900 | 0.239 | 39.956 |
| DiffLL [15] SIGGRAPH Asia'23 | 24.330 | 0.843 | 0.245 | 49.098 |
| PyDiff [65] IJCAI'23 | 25.753 | 0.897 | 0.159 | 36.263 |
| **JoReS-Diff (Ours)** | **27.347** | **0.912** | **0.121** | **27.233** |

**Table 3: Quantitative comparison on the ISTD [39].**

| Methods | ISTD | | |
|---|---|---|---|
| | PSNR ↑ | SSIM ↑ | RMSE ↓ |
| Shadow Image | 20.56 | 0.908 | 10.86 |
| DC-ShadowNet [18] ICCV'21 | 26.38 | 0.917 | 6.62 |
| EMNet [67] AAAI'22 | 29.98 | 0.940 | 5.28 |
| BMNet [66] CVPR'22 | 30.26 | 0.957 | 5.06 |
| ShadowFormer [10] AAAI'23 | 30.47 | 0.958 | 4.79 |
| LFG-Diff [33] WACV'24 | 30.64 | 0.963 | 4.93 |
| **JoReS-Diff (Ours)** | **30.89** | **0.971** | **4.66** |

**Table 4: Quantitative comparison on MIT-Adobe FiveK [38].**

| Methods | MIT-Adobe FiveK | | |
|---|---|---|---|
| | PSNR ↑ | SSIM ↑ | LPIPS ↓ |
| RetinexNet [46] BMVC'18 | 12.515 | 0.671 | 0.254 |
| KinD [62] MM'19 | 16.203 | 0.784 | 0.150 |
| Uformer [45] CVPR'22 | 21.917 | 0.871 | 0.085 |
| Restormer [57] CVPR'22 | 24.923 | 0.911 | 0.058 |
| LLFormer [41] AAAI'23 | **25.752** | 0.923 | 0.045 |
| **JoReS-Diff (Ours)** | 25.669 | **0.929** | **0.042** |

on PSNR and SSIM of 1.121 dB and 0.012, proving our capability of generalization. Although our JoReS-Diff obtains several second-best LPIPS values, it outperforms other diffusion-based LLIE methods and achieves the best PSNR, SSIM and FID on each case, which indicates the overall performance on perceptual metrics is still competitive. Consequently, the considerable performance shows the capability of suppressing noise and preserving color and details by Retinex and semantic priors, which demonstrates the effectiveness of our proposed joint condition strategy in resolving LLIE task.

As shown in Tables 3 and 4, we report the results on ISTD [39] and MIT-Adobe-FiveK [38] to evaluate the capability of other image enhancement tasks, i.e. shadow removal and exposure adjustment. We can observe the remarkable performance of our JoReS-Diff from the comparisons. To be specific, our method improves the PSNR/SSIM by 0.25 dB/0.008 than the latest method LFG-Diff, demonstrating the effectiveness of the joint prior in the shadow removal task. On FiveK [38], JoReS still outperforms previous strong baselines, such as Uformer and Restormer, and provides competitive performance comparing with LLFormer. The slight drop of PSNR may cause by the gap between the PASCAL-Context and FiveK.

## 4.3 Qualitative Evaluation

The qualitative evaluations on LOL and LOL-v2 are shown in Figs. 1, 5 and 6. As indicated by the visual comparisons, our JoReS-Diff shows superior enhancement capability and generates images with more pleasing perceptual quality. Specifically, in the first row of Fig. 5, previous methods fail to reconstruct the detailed textures of the blanket, while our JoReS-Diff provides rich details and mitigates the color gap. As for images of the middle row, although most methods enhance the white regions well, they produce incorrect color and artifacts in the regions with complex scenes. Notably, our method not only restores the print on the carton but preserves the color consistency. Furthermore, the bottom row exhibits that our JoReS-Diff is capable of recovering the vulnerable content vanishing in the results of other methods, which reasonably indicates the superior capacity of restoring naturalistic details. In Fig. 6, although PyDiff [65] shows a similar overall look to ours, the JoReS-Diff enhances the green cup and the red text with correct color and better saturation. Hence, our JoReS-Diff yields more visually pleasing results as compared to baselines, supporting our method's excellent performance in quantitative evaluation. More results are provided in supplementary material.

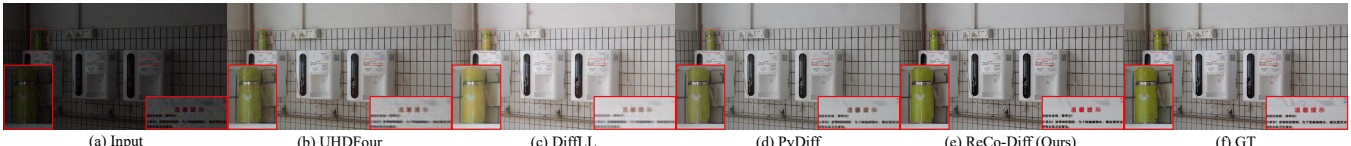

Figure 5: Visual comparison of our JoReS-Diff and the compared LLIE methods on the LOL dataset.

Figure 6: Visual comparison of our JoReS-Diff and the compared LLIE methods on the UHD-LL dataset.

## 4.4 Ablation Study

**Contribution of Retinex prior incorporation.** As shown in Table 5, we conduct experiments of several ablation settings by removing different components from the framework individually. The "w/o UNet-Cond" denotes the removal of Retinex attention layers in the UNet. Compared with all ablation settings, our full setting yields the best performance. In the case of the removal of FRCM leads to the decrease of PSNR value by an average of 0.565 dB below the baseline, which demonstrates the effectiveness of introducing the multi-scale features from ANet into the refinement process. The comparison between "w/o UNet-Cond" and the baseline shows a similar degradation level, resulting in an average reduction of 0.676 dB, which proves the necessity of integrating Retinex features into UNet. Notably, by comparing "w/o IRCM" and "w/ ALL", we

observe a significant decline (0.864 dB) that surpasses the aforementioned two cases, which serves to highlight the crucial role of our IRCM owing to its well-designed residual refinement manner. Furthermore, the comparison between all cases under "w/ ANet" and "w/o ANet" exhibits an average drop of 0.568 dB on PSNR, illustrating that the better Retinex-based condition provided by ANet eventually improves the visual quality of generated images. Under the case of "w/ ANet", the settings of "w/o SSIM loss" and "w/o Hist loss" show degradation on all metrics, demonstrating the effects of SSIM loss and Hist loss respectively. Comparing "w/ ALL" and "w/o SSIM & Hist losses" shows a favorable gain of 0.459 dB and illustrates the necessity of using the constraint of both the structure and histogram. Notably, directly multiplying the outputs of ANet produce unfavorable results since the model capacity is

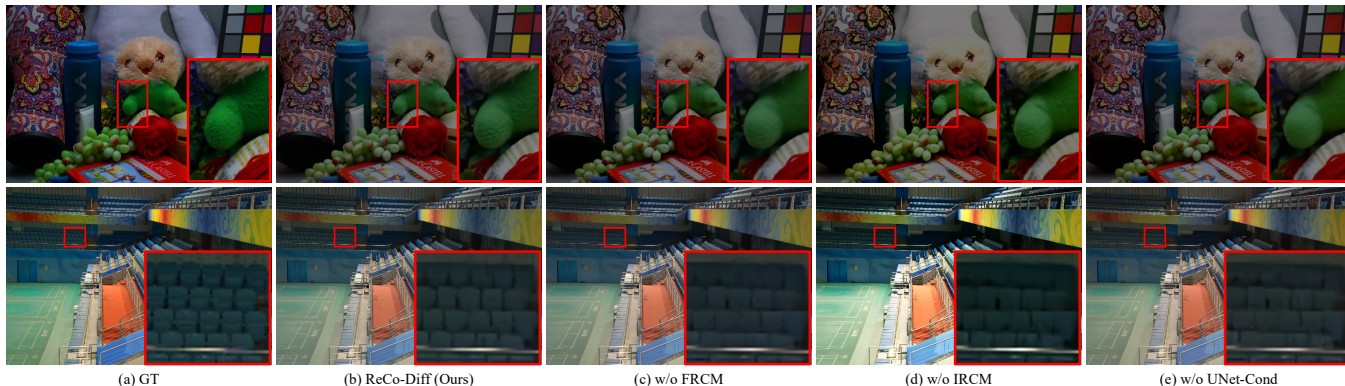

| (a) GT | (b) ReCo-Diff (Ours) | (c) w/o FRCM | (d) w/o IRCM | (e) w/o UNet-Cond |

**Figure 7: Visual comparison on the LOL/LOL-v2 datasets for investigating the contribution of key techniques of our JoReS-Diff.**

**Table 5: Ablation studies on LOL for investigating the contribution of key techniques of Retinex prior incorporation.**

| Learning | Refinement | PSNR ↑ | SSIM ↑ | LPIPS ↓ |
|---|---|---|---|---|
| | w/ ALL | 26.491 | 0.876 | 0.092 |
| | w/o FRCM | 25.834 | 0.864 | 0.133 |
| | w/o IRCM | 25.487 | 0.859 | 0.145 |
| w/ ANet | w/o UNet-Cond | 25.673 | 0.861 | 0.135 |
| | w/o SSIM loss | 26.202 | 0.866 | 0.106 |
| | w/o Hist loss | 26.194 | 0.866 | 0.105 |
| | w/o SSIM & Hist losses | 26.032 | 0.864 | 0.108 |
| | w/ ALL | 25.736 | 0.862 | 0.116 |
| | w/o FRCM | 25.264 | 0.854 | 0.149 |
| w/o ANet | w/o IRCM | 25.012 | 0.849 | 0.168 |
| | w/o UNet-Cond | 25.202 | 0.853 | 0.154 |
| ANet Output Multiplication | | 17.129 | 0.675 | 0.397 |

**Table 6: Ablation studies on LOL for investigating the contribution of key techniques of semantic prior incorporation.**

| Settings | PSNR ↑ | SSIM ↑ | LPIPS ↓ |
|---|---|---|---|
| w/ ALL | 26.491 | 0.876 | 0.092 |
| w/o CA | 26.183 | 0.865 | 0.131 |
| w/o SA | 26.024 | 0.863 | 0.118 |

similar with RetinexNet, while we can still obtain favorable results by using them as condition.

Additionally, results in Fig. 7 depict that "w/o FRCM" and "UNet-Cond" lead to the reduction of regions with detailed textures, such as uneven surfaces and closely arranged chairs. Although "w/o IRCM" preserves details better, the lack of direct guidance of Retinex components induces the color shift and unnatural illumination. Fig. 8 shows that ANet reduces the noise (top rows) and enhances the illumination (bottom rows) in early steps, and improves the quality of Retinex-based conditions in the whole process. Furthermore, we can see the effects of SSIM and Hist losses by comparing Retinex output at step 8. The reflectance and illumination maps in the green box contain more artifacts and unsatisfactory brightness distribution.
**Contribution of semantic prior incorporation.** As shown in Table 6, we conduct the ablation studies to investigate the contribution of cross-attention and self-attention in the semantic attention layers. By comparing the performance degradation caused by "w/o CA" and "w/o SA", it is conspicuous that the self-attention mainly

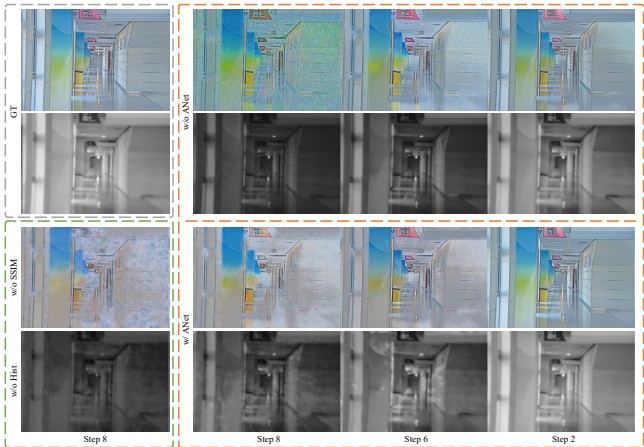

**Figure 8: Ablation study of ANet in terms of reflectance (row 1,3) and illumination (row 2,4) maps. Images in orange box shows the effects of ANet, and green box shows the effects of SSIM and Hist loss corresponding to Table 5.**

benefits on PSNR and SSIM, while the cross-attention can improve LPIPS more. The results reasonably support the motivation of designing the semantic attention layers in Section 3.3. More ablation studies are provided in supplementary material.

## 5 CONCLUSION

In this paper, we propose a diffusion model with joint Retinex and semnatic priors, JoReS-Diff, for image enhancement tasks. JoReS-Diff combines Retinex prior and diffusion model in two stages: Retinex-based condition learning and conditional refinement. In the learning stage, we utilize DNet to obtain initial decomposed maps and then provide robust Retinex prior by ANet. In the refinement stage, Retinex prior is integrated by Retinex attention layers and RCMs in RNet to control the diffusion process and produce better color and details. The semantic prior is provided by an off-the-shelf segmentation model and incorporated by semantic attention layers with both self- and cross-attention, preserving semantic consistency of the output. Extensive experiments demonstrate that our JoReS-Diff outperforms state-of-the-art methods on representative benchmarks. We also hope to encourage more works to develop DMs incorporating intrinsic characteristics of low-level vision.

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
