# OpenReview forum: "JoReS-Diff: Joint Retinex and Semantic Priors in Diffusion Model for Low-light Image Enhancement"
_acmmm.org/ACMMM/2024/Conference — MM2024 Poster_

### Official Review · Reviewer_QcuJ · 2024-05-11

**Rating:** 5
**Confidence:** 3

**Summary:**

JoReS-Diff is an approach for low-light image enhancement (LLIE) utilizing Retinex- and semantic-based priors to regulate diffusion model generation. By integrating Retinex prior through a decomposition and refinement network and incorporating semantic priors via semantic attention layers, our method achieves superior results. Experimental validation confirms the effectiveness of our approach, highlighting the importance of physical models and semantic guidance in LLIE.

**Strengths:**

The strengths of this article include:
1. The paper effectively integrates Retinex prior through a decomposition and refinement network and incorporation of semantic priors via semantic attention layers.
2. The proposed method generates images with more pleasing perceptual quality, preserving color consistency and restoring naturalistic details.
3. The paper includes ablation studies to analyze the contribution of different components in the proposed method. The results demonstrate the effectiveness of the Retinex attention layers, F/IRCMs, and IRCM in improving the performance.
4. This paper presents a well-organized layout of figures and tables, which enhances readability and comprehension.

**Limitations:**

1. How to verify the semantic consistency？ The author should conduct further analysis on the impact of the segmentation network, such as visualizing the content of the segmentation output and determining whether the enhancement results are consistent with semantics. Or it would be better to use a segmentation task to evaluate the enhanced output.
2. Computational complexity analysis: This work involves the use of multiple components and network structures, which may lead to increased computational complexity and potentially require more computational resources and time for training and inference. The author should provide the computational complexity analysis.

**Suitability:**

2

---

### Official Review · Reviewer_mgCh · 2024-05-22

**Rating:** 4
**Confidence:** 3

**Summary:**

This paper presents two improvements for diffusion models used in low-light image enhancement. Firstly, to enhance the consistency of color and content in low-light image enhancement, the paper proposes conditional generation and conditional refinement modules based on the Retinex theory. Secondly, to address the issue of unnatural textures in diffusion models, the paper employs a semantic model to constrain the generation capabilities of the diffusion model. In the experimental section, the proposed approach is compared with several state-of-the-art methods on multiple datasets, demonstrating superior performance.

**Strengths:**

1. The paper introduces a diffusion model for low-light enhancement tasks based on Retinex prior and semantic prior. It investigates the role of physical models and semantic guidance in controlling the generation capabilities of the diffusion model.
2. Quantitative experiments were conducted on multiple datasets, comparing with a wide range of state-of-the-art methods, and the proposed approach achieved the best results in the majority of the experiments.
3. In visual comparisons with multiple state-of-the-art methods, the model demonstrated its capability to preserve details and maintain color consistency.

**Limitations:**

1. In section 4.4, under the Contribution of semantic prior incorporation subsection, there is a lack of ablation studies that include the removal of both Semantic Attention (SA) and Content Attention (CA), as well as the removal of the Segmentation Network from the experiments. The inclusion of such experiments would provide  evidence to demonstrate the role of semantic information in eliminating unnatural textures.
2. In the first occurrence of a symbol, an explanation should be provided, for instance, the “**y**” in Equation 1.
3. The meaning of $\hat R$ in Equation 5 differs from that in Equation 6, with Equation 6 including an additional logarithmic operation compared to Equation 5.
4. The color distinction between the **Retinex Attention Layer** and the **Semantic Attention Layer** in Figure 3 is not clear.
5. In Figure 3, the arrow above circle $x_{t-1}$ at the bottom has an incorrect source; it should be directed from $\hat{x}^{'}_{0}$.
6. The notation for ${\hat c}_t$ in section 3.2.1 is inconsistent, with the initial representation being ${\hat c}_t=[{R}^{'}_t, {L}^{'}_t]$ and the subsequent representation being ${\hat c}_t=[{R}^{'}_t, {L}^{'}_t, {F}_t]$. It is recommended to either unify the notation or use two distinct symbols to differentiate between the two.

**Suitability:**

2

---

### Official Review · Reviewer_dyZR · 2024-05-24

**Rating:** 2
**Confidence:** 4

**Summary:**

The authors propose a low-light enhancement strategy based on the Diffusion model, which achieves good enhancement performance by fusing the retinex decomposition prior and the semantic prior.

**Strengths:**

1.The authors focus on two difficult issues in low light enhancement: color and missing details, and designs corresponding strategies to improve them.
2.The authors design comprehensive experiments and validate the effectiveness of the method on multiple datasets.

**Limitations:**

1.Experimental results. For the experimental results of Table 1 on LOL, some diffusion based methods have achieved better results, such as GlobalDiff (NIPS2023,https://paperswithcode.com/sota/low-light-image-enhancement-on-lol), but the authors did not compare them. For visual results, although the results in Figures 1 and 5 have improved compared to previous methods, there are still deviations in details and colors.
2.The role of DNet and ANet: Is it necessary to divide the prior part of the retinex into these two networks? The author explains that pretrained DNet may initially generate poor results, while randomly initialized ANet may not generate good results before training as well. I hope the author can further explain why use ANet.
3. The author seems to have used the wrong name，are ReCoDiff(in fig.5,6,7) and JoResDiff the same method?.
4.How about the computing complexity? The authors are suggested to give the comparison in terms of memory cost, FLOPS, and inference time.

**Suitability:**

2

---

### Official Review · Reviewer_NaYV · 2024-05-25

**Rating:** 4
**Confidence:** 4

**Summary:**

This paper proposes an effective low-light enhancement network based on the diffusion model, named JoReS-Diff. Different from previous studies, the proposed method injects the Retines prior obtained by a pre-trained decomposition network and semantic prior obtained by a pre-trained segmentation network into a diffusion model for producing high-quality images. It evaluates the proposed method on several benchmark datasets and demonstrates that the proposed method can achieve the best performance in terms of distortion and perceptual metrics.

**Strengths:**

1. The paper clearly points out the limitations of the existing methods and describes the motivation of the proposed method.

2. The structure of the paper is well-organized and easy to follow.

3. The proposed method has been evaluated on several benchmark datasets and achieves the best results, which are convincing.

**Limitations:**

1. The pipeline illustrated in Fig.3 is not consistent with that in Fig.2. Specifically, the input low-light image is the input of the segmentation network and is concatenated with the noise.

2. What are the motivations for designing such network structures of FRCM and IRCM? These structures seem to be heuristic designs.

3. The proposed method leverages a pre-trained segmentation network and a pre-trained Retinex network. How about the model complexity in terms of model parameters, the number of FLOPs, and the computational resources? Regarding the marginal improvements of this method on several datasets, the illustration of the computational complexity is necessary.

**Suitability:**

2

---

### Meta-Review · Area_Chair_DU31 · 2024-07-01

**Recommendation:** Accept (Poster)
**Confidence:** 5

**Metareview:**

This paper was reviewed by four experts in the field. The paper received mixed reviews BA, WR, BA, WA. The reviewers liked the idea of this paper, its thorough evaluation and clear motivation. The reviewers also raised the following concerns on the application scenarios, e.g., the computing complexity, comparisons with other methods and the module design details. Based on the rebuttal, the AC feels that most concerns have been resolved. Based on the reviews, the AC would like to recommend the acceptance of this paper, and suggest the authors to include the added experiments in the rebuttal to the final version.